# Single-Nucleotide Polymorphisms of Growth Hormone Gene and Its Relationship with Growth Traits in Black Bengal Goats

**DOI:** 10.3390/ani14060834

**Published:** 2024-03-08

**Authors:** Chollada Buranakarl, Sumonwan Chamsuwan, Sumpun Thammacharoen, Panupat Ratchakom, Natthaya Chuaypen

**Affiliations:** 1Department of Physiology, Faculty of Veterinary Science, Chulalongkorn University, Henri Dunant Rd., Pathumwan, Bangkok 10330, Thailand; sumonwan.c@chula.ac.th (S.C.); sprueksagorn@hotmail.com (S.T.); 2Chiang Rai Provincial Livestock Office, Department of Livestock Development, Chiang Rai 57000, Thailand; pratchakom@gmail.com; 3Metabolic Diseases in Gut and Urinary System Research Unit (MeDGURU), Department of Biochemistry, Faculty of Medicine, Chulalongkorn University, Henri Dunant Rd., Pathumwan, Bangkok 10330, Thailand; natthaya.ch56@gmail.com

**Keywords:** Black Bengal goat, growth hormone gene, growth traits, single-nucleotide polymorphism

## Abstract

**Simple Summary:**

Raising goats in tropical regions has become more popular in recent years. Black Bengal goats are meat-type goats that can resist hot environments. Identifying some selective genes related to growth is one of the most important factors for genetic improvement. The somatotropic axis is related to kid growth, while the growth hormone has been proven to be essential for the kid growth rate. The single-nucleotide polymorphisms of the growth hormone gene in dairy cows and many small ruminants showed that A781G and A1575G are related to growth rate. However, data on Black Bengal goats in Thailand are still limited and have shown variable results. The aims of the present study were to investigate genotypes and their association with the preweaning growth rate. Genotypes were identified from 77 goats of both sexes, and our results showed three different genootypes—AA, AB, and BB—for A781G, while only a CC genotype was found for A1575G. A relationship was found between preweaning growth and the genotypes of A781G, in which the AA genotype had the highest growth rate. Therefore, the identification of the genotypes of A781G can be applied in the management of breeding programs for Black Bengal goats in Thailand.

**Abstract:**

The single-nucleotide polymorphisms (SNPs) of the growth hormone (GH) gene could be related to growth traits, particularly in farm animals. This study aimed to identify the SNPs of the *GH* gene (A781G and A1575G) in Black Bengal (BB) goats in Thailand. Seventy-seven BB goats of both sexes were recruited, and their genotypes were identified. Preweaning growth at birth (weight, W0; height, H0; length, L0; and chest girth, C0) and at 10 weeks postpartum (W10, H10, L10, and C10), including average daily gain (ADG) at 0–4 weeks (ADG0–4W), 4–8 weeks (ADG4–8W), and 8–12 weeks (ADG8–12W), was compared among the different genotypes in goats born from twin litter-size dams. The results showed one genotype, CC, for A1575G and three genotypes, AA, AB, and BB, for A781G. The AA gene had significantly higher W10 than AB (*p* < 0.05) and BB (*p* < 0.05). The AA had significantly higher L10 than AB (*p* < 0.05), while C10 was only higher in male goats (*p* < 0.01). The ADG4–8W of the AA genotype was significantly higher than the BB genotype (*p* < 0.01). We came to the conclusion that A781G is associated with growth traits during the preweaning period, while the AA genotype showed better performance than the other genotypes.

## 1. Introduction

Black Bengal (BB) goats are small-to medium-sized meat-type goats characterized by early maturation and the ability to produce many kids in one litter. Most of them are black in color and have short legs, little horns, and tight bodies. This breed is a type of goat that produces high-quality meat, milk, and skin. Moreover, Black Bengal goats have the characteristics of early maturation and a high resistance to tropical environments and disease [1]. Thus, they hold economic value, especially for farmers living according to the sustainable development theory. BB goats were imported from Bangladesh to Thailand in 2005 and raised under the supervision of the Chaipattana Foundation in the Chiang Rai province in the northern part of Thailand. This farm hosts the only BB purebreds raised in Thailand. This farm was set up to conserve and improve the characteristics of purebred BB goats and distribute the goats to local farmers.

In order to breed a purebred BB goat, the high growth rate and survival of kids are essential aspects to consider. In ruminants, growth and development are affected by many factors. One of the most important factors is the growth hormone (GH), which is one of the most important hormones as it regulates growth in every species. The growth hormone is synthesized in the anterior pituitary gland and released into the bloodstream. The GH has direct or indirect effects on postnatal somatic growth, as well as anabolic processes such as protein synthesis, in many tissues. The hormone regulates lipid metabolism, carbohydrate metabolism, and somatic growth. This hormone plays a crucial role in animal production, especially in farm animals. The hormone has previously been reported to impact both milk and meat production [2,3,4]. The activity of the GH is mediated through insulin-like growth factor-1 (IGF-1), which is synthesized in the liver. The concentration of IGF-1 in the colostrum of BB goats was high but subsequently declined to low levels in the milk of BB goats [5]. Kids consuming colostrum from a dam with high concentrations of IGF-1 showed high growth rates [6]. Many studies have investigated the roles of SNPs of the *GH* gene and their involvement in growth, reproductive performance, milk yield, and carcass quality in ruminants.

In cattle, the growth hormone gene is encoded by 1800 base pairs (bp) consisting of five exons separated by four intervening sequences [7]. Changes in the GH’s polymorphisms could affect growth hormone concentration and some metabolites. A study on Japanese Black heifers showed that the SNPs of the *GH* gene could alter the GH concentration in response to a growth hormone-releasing injection, as well as that of other important hormones and metabolites such as insulin, plasma IGF-1, leptin, and glucose [8]. Alleles that differ by amino acid mutations affect the carcass weight and beef marbling score [9], stress responses [10], birth weight, and weight at 30 days of age [11].

In goats, variations in the nucleotide sequences of the five exons of the *GH* gene have been identified using single-strand conformation polymorphism (SSCP) techniques [12]. Among these genetic variations, the A781G (c.781A>G) and A1575G (c.1575A>G) of the *GH* gene have been widely investigated. The A781G is located at exon 2 of the *GH* gene and causes an amino acid change from serine to glycine, while the A1575G is located at exon 4 and has no influence on the coding potential [13]. Due to the effect of the GH on growth development and lactation, several studies have observed a correlation between the SNPs of *GH* genes and milk yield performance using SSCP and polymerase chain reaction single-strand conformation polymorphism (PCR-SSCP) techniques [12,14,15]. Few studies have focused on the association between GH polymorphisms and growth traits. In a study on Boer goat bucks, subjects with the AB genotype of A781G were found to have significantly higher birth chest girth and weaning weight values compared with those with the AA genotype, while animals with the CC genotype of A1575G had greater weaning height values compared with those with the CD genotype [13]. On the other hand, there were no differences in body size between those with genotype AA of A781G and those with genotype AB of A781G in Kacang goats in [16]. A study on Sabural goats showed that growth performance parameters such as weight, height, body length, and chest circumference were higher at birth in those with genotype AB of A781G than those with genotype AA [17].

In a study on BB goats, the SNPs of the *GH* gene were identified using SSCP techniques, and there were seven and five haplotypes in exons 4 and 5, respectively [18]. Moreover, a study on the 245 bp fragments (partial intron 1, exon 2, and partial intron 2) of the *GH* gene polymorphisms in BB goats revealed five genotypes (AA, AB, AC, AD, and CC), and those with the AC genotype had greater birth weight compared to those with the CC genotype [19]. Another study, specifically that of Dayal (2016) [20], showed that the 472 bp fragment genotypes (partial intron 2, exon 3, intron 3, and partial exon 4), but not the 245 bp fragment genotype, had a significant effect on body weight at 6 and 9 months of age in BB goats. However, the associations between A781G or A1575G of the *GH* gene and the growth traits in BB goats have not been explored yet.

This study aimed to investigate growth hormone gene SNPs in BB goats that were raised in Chiang Rai province, Thailand. The SNPs of the *GH* gene (A781G and A1575G) were identified using DNA extracted from hair, and the growth rate was investigated 3 months postpartum, along with IGF-1 and leptin concentrations. The information presented herein on the SNPs of the *GH* gene and their association with growth rate will provide useful insights for genetic selection in farms to conserve the good characteristics of BB goats.

## 2. Materials and Methods

The study protocol was approved by IACUC committee, Faculty of Veterinary Science, Chulalongkorn University (protocol number 2231055). The study sample consisted of a total of 77 goats (44 mature females, 5 mature males, 16 young females, and 12 young males), aged between 7.1 and 68.4 months old. Data including buck and dam identity and size at birth were recorded. The weight and conformation of all goats were also recorded. The SNPs of the *GH* gene (A781G and A1575G) were identified by extracting DNA from hairs collected from the sample goats.

### 2.1. Animal Housing and Management

The purebred Black Bengal goats were originally imported from Bangladesh, along with known pedigree certificates. All the goats were kept and raised at the farm dedicated to the Chaipattana Foundation’s Black Bengal Goat Domestication Project located in the Chiang Rai province in the northern part of Thailand. The colony of these goats was preserved within the project by using a specific breeding line. The population of offsprings was controlled through selection and programs distributed from the project to farmers. These management strategies were performed to ensure the qualitative and quantitative maintenance of purebreds and improve the genetic selection outcomes. All goats were housed in a conventional open housing system, with 2–3 goats per pen (3 × 3 m). All goats were mated and delivered naturally. After delivery, kids were allowed to stay and receive both colostrum and milk with dams until weaning at approximately 12 weeks. All goats were fed twice a day at 07.00 and 18.00 and had free access to water. The goats’ diets consisted of commercial concentrate at approximately 200 g/goat/day and roughage including 2 kg/goat/day of Napier grass and 1 kg/goat/day of Pangola grass. The compositions of the diet regimens are shown in Table 1. The newborn kids remained with their mothers and were allowed to consume colostrum and milk. At approximately one month of age, the kid goats were provided with a concentrate mixture while they were raised by their mothers. A vaccination program against foot-and-mouth disease virus as well as serum titer injections against caprine arthritis encephalitis virus and brucellosis were performed yearly in all goats. Endoparasites and ectoparasites were routinely eradicated.

The goats in this study consisted of pregnant dams with a known preweaning growth rate and their offsprings, for which growth traits were recorded after delivery. The SNPs of the *GH* gene were obtained from DNA extracted from the hair of each goat.

### 2.2. Measurement of Body Conformation

General information on the dams and bucks, such as identity, age, sex, litter size when they were born, and growth traits, was obtained from farm records, and we also obtained information on kids born between March 2002 and July 2002 from these dams and bucks. Characteristics such as body weight, length, height at the withers, and chest girth were assessed for each individual goat, and the goats were separated from their mothers 1 h before weighing. Body weight was measured by placing a kid inside a basket and weighing them using electronic digital scales (CRD 30, CST Instruments, Bangkok, Thailand) every week after birth until 12 weeks of age. Conformation at birth, measured in terms of the weight (W0), length (L0), height (H0), and chest girth (C0) of the newborn goats, and at 10 weeks postpartum (W10, L10, H10, and C10), was measured while the kids were in a standing position. Body length was measured from the point of the shoulder to the pin bone. Height was measured from the ground at the front hoof to the point of the withers. Chest girth was measured as the body circumference from the heart, to just behind the elbows, and up to the withers. Average daily weight gain (ADG) was calculated at 0–4 (ADG0–4W), 4–8 (ADG4–8W), and 8–12 (ADG8–12W) weeks of age.

### 2.3. Hair Collection

From each goat, about 30–40 hair shafts with attached follicles were pulled out from the buttock near the tail using alligator forceps and collected in plastic bags for DNA extraction.

### 2.4. Analytical Procedures

#### 2.4.1. Sample Preparation for the Genotyping of the GH Polymorphisms

Approximately 20–30 plucked hairs from each goat were cleaned with deionized water, followed by rinsing with 100% ethanol. The hairs were dried for about 10 min, and a bulb of hair from each shaft was collected (not longer than 5 mm) using a sterilized scalpel or scissors.

#### 2.4.2. DNA Extraction and Collection

The DNA from the hair follicles was isolated using the DNeasy blood and Tissue kit (Qiagen, Hilden, Germany). DNA concentration and purification was carried out using a microvolume spectrophotometer (DeNovix Inc., Wilmington, DE, USA). The purified DNA was kept at −20 °C until further analysis.

The primers and probes used for the TaqMan real-time PCR assay of A781G and for the sequencing of A1575G are shown in Table 2.

#### 2.4.3. TaqMan Allelic Discrimination Assay for A781G Detection

Genotypic analysis of the *GH* gene polymorphisms (A781G) was performed using the TaqMan real-time PCR assay (Thermo Fisher Scientific, Waltham, MA, USA). The intensities of each probe were plotted on an allelic discrimination graph. Positive and negative controls were also included in each experiment to validate the assay’s accuracy. The thermal cycling conditions included an initial step at 95 °C for 10 min, 40 cycles of denaturation at 95 °C for 15 s, primer annealing at 60 °C for 60 s, and extension at 60 °C for 30 s, followed by a final extension step at 72 °C for 2 min. Fluorescent signals from VIC and FAM were obtained at the end of each cycle. The allelic discrimination plot was analyzed using QuantStudio^TM^ Design & Analysis Software v1.5.2 (Thermo Fisher Scientific, Waltham, MA, USA).

#### 2.4.4. Nucleotide Sequencing Analysis for A1575G Detection

The PCR was performed in a 25 μL mixture containing 10 μM primers, 5 mM dNTP (deoxyribonucleotide triphosphate), 2.5 μL 10× reaction buffer, 1 unit of Taq-DNA polymerase (Yeastern Biotech, Taipei, Taiwan), and 50 ng genomic DNA as template. The PCR cycling conditions were predenaturation at 95 °C for 1 min, followed by 35 cycles of denaturation at 95 °C for 30 s, annealing at 52 °C for 30 s, and extension at 72 °C for 45 s. The amplified products were analyzed using 2% agarose gel electrophoresis and visualized by ethidium bromide staining. A sequencing library was constructed using a Celemics NGS Library Preparation Kit (Celemics, Seoul, Republic of Korea). Next-generation sequencing was performed using the Illumina HiSeq. 2500 platform (Illumina, San Diego, CA, USA) using the MiSeq™ Reagent Kit v3. Sequence reads in FASTQ format were aligned to The National Center for Biotechnology Information (NCBI) databases using the Basic Local Alignment Search Tool (BLAST).

### 2.5. Statistical Analysis

All analyses were performed using the SAS program version 9.4. Data are presented as mean ± SEM. Descriptive statistical analyses were performed. Only dams with two kids per litter were used to calculate the effect of genotype and sex on kid growth traits using the general linear model (GLM) of SAS. The final model included the genotypes (AA, AB, and BB) and sex (male and female) and their interaction. *p*-values less than 0.05 were considered to show statistical significance, while *p*-values between 0.05 and 0.10 were considered to show tendency.

## 3. Results

### 3.1. Characteristics of Goats

A total of 60 female and 17 male goats was recruited for this study. The average age at blood collection was 25.3 ± 2.1 months old (range 7.1–68.4 months). The dam parity number and litter size were 2.50 ± 0.25 (range 1–7) and 1.71 ± 0.10 (range 1–3), respectively, for the relationship study.

### 3.2. Genotypes of Growth Hormone Gene A781G and A1575G

The genotypes and allele frequencies of A781G are shown in Table 3. Three types of genotypes were found: AA, AB, and BB (Figure 1). The highest percentage of 55% was found for the heterozygous genotype (AB), while the lowest percentage of 19% was found for BB. Among the dams, bucks, and kids, our genotypic analysis of A1575G revealed only the CC genotype (Table 3, Figure 2).

### 3.3. Effects of Genotype and Sex on Growth Traits in Twin Litter-Size Goats

To avoid the effects of litter size on growth traits, only goats that were born from twin litters were used (*n* = 56). Only sex and genotype affected the growth traits. The effects of the genotypes on growth traits are shown in Table 4. Kids with the AA genotype had significantly higher W10 than those with AB (*p* = 0.025) and BB (*p* = 0.046). The L10 in AA was higher than in AB (*p* = 0.034) and tended to be higher than in BB (*p* = 0.085). The C10 of the AA genotype was higher than that of the AB genotype in only the male goats (46.5 ± 1.4 vs. 41.8 ± 0.8 cm, *p* < 0.009). The ADG4–8W of the AA genotype was the highest (73.7± 5.8 g/day), being significantly higher than that of the BB genotype (44.6 ± 6.8 g/day, *p* = 0.006), tending to be higher than in the AB genotype (56.7 ± 5.0 g/day, *p* = 0.078). No differences in ADG between the AB and BB genotypes were observed (*p* = 0.330).

The effects of sex, genotype, and their interaction on the growth traits of the goats are shown in Table 4. Male kids had higher H0, L0, C0, and H10 than female kids (*p* < 0.05).

## 4. Discussion

Polymorphisms of the growth hormone gene were successfully identified in the present study using hair follicles, as was the case in a previous study on Laker goats [21]. This study is the first to identify polymorphisms of the *GH* gene located at exon 2 (A781G) and exon 4 (A1575G) in BB goats. The substitution of a guanine (G) at position 781 causes an amino acid change from serine to glycine, while the A1575G does not result in a change in amino acid [13]. The SNPs of the *GH* gene in BB goats have been studied using different techniques, such as SSCP techniques, and an association with birth weight has been demonstrated [18,19,20].

The A781G polymorphism of the *GH* gene is one of the most interesting SNPs, and it has previously been studied in ruminants and associated with growth rate. In the present study, three different genotypes, AA, AB, and BB, were found for the single-nucleotide polymorphism A781G in BB goats in Thailand. This is the first study to elucidate the BB genotype in Black Bengal goats, as an absence of this homozygous mutant (BB) has been reported in many studies on both goats and Indian sheep for the “lethal” SNP (A781G) of the growth hormone gene [22]. Previous studies have failed to demonstrate the homozygous BB genotype in Boer goats [13] and Motou and Boer goat breeds [23]. It has been concluded that the absence of BB may be due to natural selection, and the BB genotype is a recessive lethal gene. Nevertheless, in Gaddi goats of the Western Himalayas in India, a polymorphism of *GH1* located at exons 2 and 3 showed three different genotypes—AA, AB, and BB—with frequencies of 0.27, 0.52, and 0.22, respectively [24]. The results of this study are comparable with the present study, in which the frequencies of AA, AB, and BB were 0.260, 0.545, and 0.195, respectively. Previous studies on many breeds of goats have found only AA and AB genotypes with different gene frequencies. The AB genotype has reportedly been found in different goat breeds with high frequencies of 0.84 in Boer goats [13], 0.82 in Sirohi goats, 0.90 in Barbari goats [25], 0.90 in Kacang goats [16], 0.65 in Saburi goats [17], 0.95 in Barki goats, 0.90 in Damascus goats, 0.95 in Zaraiba goat breeds [26], 0.76 in crossbred Anglo-Nubian dairy goats [27], 0.94 in Laker goats [21], and 0.84 in Matou goats [23].

The allele frequencies for A and B were 0.532 and 0.468, respectively, similar to the values found in a previous study in Gaddi goats (0.53 for A and 0.47 for B) [24]. The allele frequencies for A and B were 0.581 and 0.419 in Boer goats [13], 0.59 and 0.41 in Sirohi goats, 0.55 and 0.45 in Barbari goats [25], 0.547 and 0.452 in Kacang goats [16], 0.68 and 0.32 in Saburai goats [17], 0.47 and 0.53 in Barki goats, 0.45 and 0.55 in Damascus goats, 0.47 and 0.53 in Zaraibi goats [26], 0.62 and 0.38 in crossbred Anglo-Nubian dairy goats [27], and 0.547 and 0.452 in Laker goats [21]. The goat population in this study was not in the Hardy–Weinberg equilibrium (*p* < 0.05) for the A781G of the *GH* gene, which might be due to population selection. Another SNP that may be involved in growth traits is A1575G. For this SNP, we only identified one genotype, CC. This result differs from those of earlier studies in which only the heterozygous CD genotype was found in Boer goat bucks [13] and Saburai goats [17].

The birth weight and average daily weight gain values of the goats in this study were within an acceptable range, and they are affected by several factors, including genetic factors, diet, season, sex of kids, litter size, and parity number [1]. Litter size had a major effect on kids’ growth rates, especially in the Black Bengal goats in which a large population of dams produced twin kids. A previous study showed that kids born from dams with a high litter size had lower ADG up until 9 weeks postpartum compared to those born from dams with a low litter size [28]. Therefore, only goats born from twin litters (*n* = 56) were used in the present study. Considering the effects of the genotypes on kid growth, the AA genotype exerted a greater effect on weight, length, and chest girth at week 10 compared to the AB genotype. Subjects with the AA genotype also tended to have higher weight, length, and chest girth at week 10 than those with the BB genotype. In terms of ADG, the AA genotype had the highest ADG4–8W, with a value significantly higher than that of the BB genotype.

Our results show that the growth rate associated with the AA genotype was superior to the growth rates associated with both the AB and BB genotypes. These results are not consistent with those of previous studies on goats. In Gaddy goats in which three different genotypes were identified, those with the BB genotype had higher body weight at 9 months compared to those with the AB genotype [24]. In a study on female Saburai goats, both at birth and at weaning, those with the AA genotype showed worse performances in terms of parameters such as weight, chest girth, length, and height, compared with those with the AB genotype [17]. In a study on Boer bucks, genotypic differences had only a minor effect on growth performance at birth, preweaning (80 days), and 11 months of age, although at-birth chest girth and weaning weight were lower in those with the AA genotype than those with the AB genotype [13]. In studying Sirohi and Barbari goats, Singh and colleagues (2015) [25] found that the AB genotype is superior to the BB in terms of its effect on chest girth at all ages, although the difference between the two genotypes was not significant. Nevertheless, one study on adult female Kacang goats showed that the A781G polymorphism had no effect on height, chest girth, and length [16]. The discrepancies between genotypes and indicators in different breeds may be due to other genes that may influence the expression of the *GH* gene, population selection, and the time of growth rate measurement. In the present study, the lowest growth rate at the preweaning stage was demonstrated in Black Bengal goats with the BB genotype. Further research is needed to elucidate the BB genotype of A781G of the *GH* gene in other goat breeds from Southeast Asia. However, the differences in growth rate in the present study cannot be attributed to differences in nutrition or environmental factors, since all the goats in this study were subjected to the same feeding strategy and management conditions.

In addition to growth performance, the A781G SNP of the *GH* gene is also related to other performance parameters, such as milk yield. In a study on crossbred Anglo-Nubian dairy goats, goats with the AB genotype tended to have higher milk yields than those with the AA genotype, although the difference between these two groups was not significant [27]. It has been suggested that the GH is crucial for the maintenance of lactation and milk secretion via increased nutrient utilization and mammary blood flow. Moreover, in another study on Matou goats, *GH* genotypes also affected litter size, as goats with the AB genotype were found to have higher litter sizes than those with the AA genotype, and combination with A1575G resulted in strong effects [23]. The number of corpora lutea and ova harvested was also higher in those with the AB genotype compared to those with the AA genotype.

Using sequencing analysis, the present study identified only the CC genotype for A1575G of the *GH* gene. A heterozygous genotype was not detected, which differs from a previous study in which only the CD heterozygous genotype was found in Saburai goats [17]. Two genotypes, CC and CD, have been found in Boer bucks [13,23] and Matou goats [23]. These genotypes have been found to have little effect on growth traits and birth weight [13]. Moreover, A781G and A1575G had a strong combined effect on growth performance in a study in which subjects with the AACD genotype had the lowest weight, length, height, chest girth at birth, weaning weight, preweaning gain, and 11-month weight values. A study on Gaddi goat showed two different genotypes, AB and BB, with frequencies of 0.24 and 0.76, respectively [24]. Subjects with the AB genotype had greater heart girth than those with the BB genotype at 9 and 12 months of age. Nevertheless, our results show a monomorphic genotype for A1575G. The variation in our results may be due to the limited number of goats featured in the study.

In addition to growth traits, A1575G was also related to litter size in a study on Matou goats in which subjects with the CC genotype had higher litter sizes than those with the CD genotype, and the effect was remarkable when it was combined with A781G [23]. This gene also affected the number of corpora lutea in the goats.

In the present study, sex also had an effect on growth traits, especially at birth (H0, L0, C0, and H10): the males were bigger than the females from birth until weaning. Our data are similar to those in a previous report on BB goats in Thailand [28].

This study has limitations. Since the Chaipattana Foundation’s Bengal Goat Domestication Project’s Farm was the only one farm that raises and conserves BB goats featured in this study, the number of goats in this study was limited, meaning that we had a small sample size for determining the association with growth rate.

## 5. Conclusions

In conclusion, we identified three genotypes of the A781G SNP of the *GH* gene—AA, AB, and BB—while only the homozygous CC genotype was found for A1575G. The A781G SNP of the *GH* gene showed effects on growth rate at 4–8 weeks postpartum, as the growth traits of the subjects with the AA genotype were superior to those with the AB and BB genotypes. The results of this study may be applied to allow for the use of the A781G SNP of the *GH* gene in genetic selection to improve the growth performances of Black Bengal goats.

## Figures and Tables

**Figure 1 animals-14-00834-f001:**
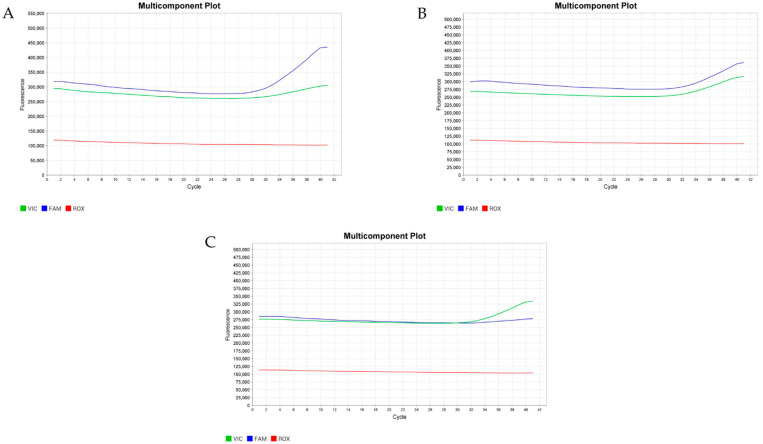
Real-time amplification for the A781G of the *GH* gene TaqMan assay. The blue curve indicates amplification from the “A” allele, while the green curve represents the “G” allele. The pictures show the predicted amplification curves for the AA genotype (**A**), AB genotype (**B**), and BB genotype (**C**).

**Figure 2 animals-14-00834-f002:**
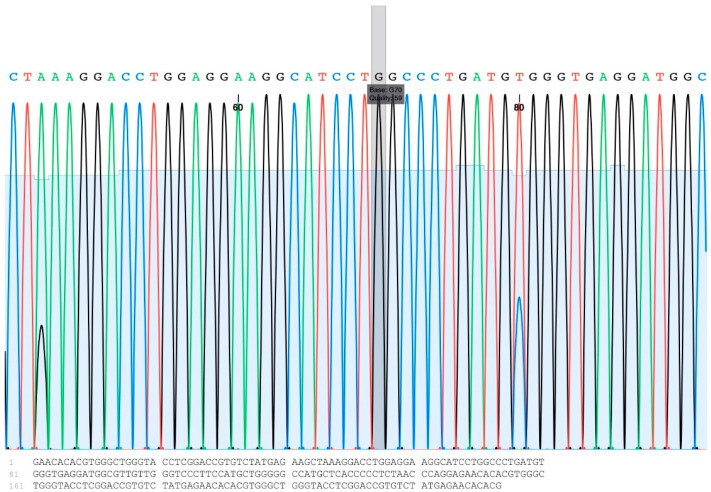
An example of an A1575G DNA sequencing chromatogram showing the “G” base at the target position (the gray box).

**Table 1 animals-14-00834-t001:** The chemical compositions of dams’ diets (DM basis).

	Concentrate	Napier Grass	Pangola Hay
Dry Matter	91.15	16.22	92.77
Organic Matter	85.73	88.72	94.58
Ash	14.27	11.28	5.42
Crude Protein	17.24	15.04	3.54
Crude Fat	2.92	1.85	1.21
Crude Fiber	12.58	30.76	30.59
NDF	34.06	61.20	66.24
ADF	16.35	39.30	37.58

NDF: neutral detergent fiber; ADF: acid detergent fiber.

**Table 2 animals-14-00834-t002:** The names and sequences of the *GH* gene primers, probes, and their product sizes.

*GH* Gene Polymorphisms	Sequences	Product Size
A781G		88
Forward	GAAGGTGTCAGCAGCCAGTTG	
Reverse	CGCCTTCCCAGCCATGT	
Probe 1 (A)	CAAACAGGCTGGACAA	
Probe 2 (B)	AAACAGGCCGGACAA	
A1575G		151
Forward	CCCAGCCCACGTGTGTTC	
Reverse	ACCTCGGACGTGTCTATGAGA	

**Table 3 animals-14-00834-t003:** Genotypes and allele frequencies of growth hormone gene A781G and A1575G.

Gene	N	Genotype	Allele
Gene A781G	77	AA 20 (0.26)	A 82 (0.53)
		AB 42 (0.55)	B 72 (0.47)
		BB 15 (0.19)	
Gene A1575G	77	CC 77 (1.0)	C 154 (1.0)

**Table 4 animals-14-00834-t004:** Effects of genotype and sex on growth traits in twin litter-size goats.

Growth Traits	Genotype			SEM	*p*	Sex		SEM	*p*	Gene × Sex
	AA	AB	BB			F	M			
W0	1.54	1.41	1.41	0.10	0.432	1.37	1.53	0.08	0.103	0.659
H0	29.4	27.7	27.4	1.0	0.179	27.1	29.2	0.8	0.028	0.439
L0	27.7	26.3	26.9	0.9	0.394	25.8	28.1	0.8	0.018	0.327
C0	25.9	25.1	25.5	0.6	0.488	24.9	26.1	0.5	0.033	0.512
W10	7.08 ^a^	5.78 ^b^	5.65 ^b^	0.45	0.017	5.88	6.46	0.38	0.194	0.110
H10	43.6	40.7	41.10	1.2	0.097	40.4	43.3	1.1	0.021	0.495
L10	47.6 ^a^	43.6 ^b^	43.5 ^ab^	1.4	0.029	44.5	45.3	1.2	0.556	0.211
C10	44.1 ^a^	41.2 ^b^	42.1 ^ab^	1.0	0.028	41.8	43.1	0.8	0.185	0.029
ADG0–4W	84.7	77.1	75.6	7.2	0.560	75.4	82.90	6.3	0.304	0.170
ADG4–8W	73.7 ^a^	56.7 ^ab^	44.6 ^b^	6.8	0.007	57.4	59.2	5.9	0.793	0.073
ADG8–12W	70.5	51.3	53.8	10.4	0.159	51.6	65.54	8.5	0.154	0.635

Data are presented as mean; SEM = standard error of mean; ^a,b^ different superscript means statistically significant among genotypes.

## Data Availability

The data presented in this study are available on request from the corresponding author. The data are not publicly available due to permission from the Chaipattana Foundation.

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
