# Peer review of "Single-Nucleotide Polymorphisms of Growth Hormone Gene and Its Relationship with Growth Traits in Black Bengal Goats"

_animals, 2024, doi:10.3390/ani14060834_

Round 1

Reviewer 1 Report

Comments and Suggestions for Authors

The manuscript is devoted to the study of the effect of SNP in the somatotropin gene on the meat productivity of Bengal goats. The experiment was well planned. All necessary research procedures were followed, factors influencing the research error were excluded.

During the study, the authors identified single nucleotide polymorphism A781G in Bengal goats and determined its relationship with growth rate. In the discussion, the authors point out that in different goat breeds the same genotypes for the A781G polymorphism have different effects on the manifestation of the studied trait - meat productivity (weight, height, body length and chest circumference, growth rate). However, the conclusion that this is due to conditions in different goat-raising regions is a bit questionable. There is no confirmation from the scientific literature. It is also necessary to analyze the scientific literature on the influence of other genes on the expression of the somatotropin gene. Not only does this gene provide an increase in live weight, it interacts with other genes. This may be the reason for the discrepancy between genotypes and indicators in different breeds.

The authors also argue that Hardy-Weinberg population disequilibrium is due to genetic drift caused by goats originating from different geographic regions. In this case, the research materials should have indicated that the population is heterogeneous and contains animals brought from other regions. Most likely, the shift in equilibrium in the population is influenced by selection pressure, since the most productive animals are selected. These points need to be reworked a little.

Author Response

Reviewer #1

The manuscript is devoted to the study of the effect of SNP in the somatotropin gene on the meat productivity of Bengal goats. The experiment was well planned. All necessary research procedures were followed, factors influencing the research error were excluded.

Thank you very much for your valuable suggestions which are important for this study.

- During the study, the authors identified single nucleotide polymorphism A781G in Bengal goats and determined its relationship with growth rate. In the discussion, the authors point out that in different goat breeds the same genotypes for the A781G polymorphism have different effects on the manifestation of the studied trait - meat productivity (weight, height, body length and chest circumference, growth rate). However, the conclusion that this is due to conditions in different goat-raising regions is a bit questionable. There is no confirmation from the scientific literature. It is also necessary to analyze the scientific literature on the influence of other genes on the expression of the somatotropin gene. Not only does this gene provide an increase in live weight, it interacts with other genes. This may be the reason for the discrepancy between genotypes and indicators in different breeds.

Answer- I revised this part in discussion.

The authors also argue that Hardy-Weinberg population disequilibrium is due to genetic drift caused by goats originating from different geographic regions. In this case, the research materials should have indicated that the population is heterogeneous and contains animals brought from other regions. Most likely, the shift in equilibrium in the population is influenced by selection pressure, since the most productive animals are selected. These points need to be reworked a little.

Answer- We put on the source of Black Bengal goats that was imported into the farm from Bangladesh in materials and method section.

Reviewer 2 Report

Comments and Suggestions for Authors

Manuscript ID: animals-2885189

Title: Single nucleotide polymorphism of growth hormone gene and its relationship with growth traits in Black Bengal goats

"The work is a good attempt at investigating genotypes and their association with preweaning growth rate in Black Bengal goats of Thailand. For this, the authors studied two SNPs of the GH gene (A781G and A1575G) in seventy-seven goats of both sexes. The results showed three different genotypes, AA, AG, and GG for A781G, while only the GG genotype was found for A1575G. A relationship was found between preweaning growth and the A781G genotype, with the AA genotype exhibiting the highest growth rate.

The work is ACCEPTABLE. However, significant revisions are required before it can be accepted for publication. Moderate editing of English language required.

Specific Comments

1 – In the line 31 and 262, the word “borne” is written incorrectly, I believe it should be “born”.

2 – In the line 46-47, could you improve the sentence to make it clearer or more effective? Something like: “The purpose is to set up a program to conserve and improve the pure breed BB and distribute it to 46 local farmers" or “The purpose was to establish a program to conserve and improve the pure breed BB and distribute it to 46 local farmers."?

3 – In the line 55, could you improve the sentence to make it clearer or more effective? Something like: “The hormone regulates the metabolism of lipids, carbohydrates, and somatic growth."

4 – In the line 57, is it possible to change the word “involve” to “improve” or “affect”?

5 – In the line 58, could you improve the sentence to make it clearer or more effective? Something like: The action of GH is mediated through insulin-like growth factor-1 (IGF-1), which is synthesized by the liver."?

6 – In the line 60, could you improve the sentence to make it clearer or more effective? Something like: “Kids consuming colostrum from dams with high concentrations of IGF-1 had a high growth rate”

7 – In the line 66-69, could you improve the sentence to make it clearer or more effective? Something like: “A study in Japanese Black heifers showed that SNPs of the GH gene could alter the GH concentration in response to growth hormone-releasing hormone injection, as well as the concentrations of other important hormones and metabolites such as insulin, plasma IGF-1, leptin, and glucose”?

8 – In the line 79 to 82, there is a lack of standardization when comparing your work regarding the genotyping codes. In the paper, nucleotide bases are used to name the genotypes, whereas the genotypes from other articles use the letters A and B. This sometimes makes confusion.

9- In the line 72 to 95, significant portion of the text should be on Discussion.

10- In the line 104 to 106, what do the values in the parentheses mean?

11- In the line 107, the sentence “The weight and con formation of all goats” look incomplete.

12- In the Material and Methods section I got a bit confused about why using data from dams and bucks and their offspring for the experiment. Why not using only the data from offspring? Did the dams and sires' mothers receive the same treatment? Why only use data from dam with two litter size kids, since the data from single births can be corrected in the model? Also, in the line 216 says that only data from 56 goats were used, how many males and females were used?

13 – In Table 2, it may be convenient to add the Taqman probes for each SNP and theirs fluorescent signals. Please correct the name of A1575G in the right column.

14 - In my opinion, all figures in Figure 1 could be removed. They do not add any new information.

15 – In the line 209 to 214, in my opinion, section 3.3 is not necessary since it has already been demonstrated in section 3.2 that there is no polymorphism for A1575G. It is sufficient to add this information to section 3.2."

16 –In the line 217, could you improve the sentence to make it clearer or more effective? Something like: “Only sex and genes affected the growth traits.”?

17 – In the line 246, the word “Boar” is written incorrectly, I believe it should be “Boer”. It is repeated several times in the article.

18 – In the line 251, there are something strange in the text (Error! Bookmark not defined).

19 –In lines 250 to 251, would it be possible to synthesize the text into a table and include the experiment's data as well? Maybe it will be more helpful to visualize and understand the numbers and compare them

20 – In the table 4, there are something strange in the statistical results to L10.

AA – 47.6a

AG – 43.6b

GG – 43.5ab

In this case, if different superscripts means statistical significance among genotypes and how is it shown there is no difference between all three genotypes.

21 – In lines 292 to 294, it is stated that there was a sex effect on growth, but in Table 4, these differences were not statistically significant."

Comments on the Quality of English Language

In review report

Author Response

Reviewer#2

Specific Comments

Thank you very much for your kind suggestions on English and some point in gene study. I correct most of them as follows,

1 – In the line 31 and 262, the word “borne” is written incorrectly, I believe it should be “born”.

Answer-It has been English edited by MDPI already.

2 – In the line 46-47, could you improve the sentence to make it clearer or more effective? Something like: “The purpose is to set up a program to conserve and improve the pure breed BB and distribute it to 46 local farmers" or “The purpose was to establish a program to conserve and improve the pure breed BB and distribute it to 46 local farmers."?

Answer: The purpose of “conserve and distribution of goats” is for Chaipattana goat farm project which distributes goat all over country by Her Royal Highness Princess Maha Chakri Sirindhorn as part of Royal project. We study using 46 dams from this farm (not distribute to 46 farms).

3 – In the line 55, could you improve the sentence to make it clearer or more effective? Something like: “The hormone regulates the metabolism of lipids, carbohydrates, and somatic growth."

Answer-It has been English edited by MDPI already.

4 – In the line 57, is it possible to change the word “involve” to “improve” or “affect”?

Answer-It has been English edited by MDPI to “impact” already.

5 – In the line 58, could you improve the sentence to make it clearer or more effective? Something like: The action of GH is mediated through insulin-like growth factor-1 (IGF-1), which is synthesized by the liver."?

Answer-It has been English edited by MDPI already.

6 – In the line 60, could you improve the sentence to make it clearer or more effective? Something like: “Kids consuming colostrum from dams with high concentrations of IGF-1 had a high growth rate”

Answer-It has been English edited by MDPI already.

7 – In the line 66-69, could you improve the sentence to make it clearer or more effective? Something like: “A study in Japanese Black heifers showed that SNPs of the GH gene could alter the GH concentration in response to growth hormone-releasing hormone injection, as well as the concentrations of other important hormones and metabolites such as insulin, plasma IGF-1, leptin, and glucose”?

Answer-It has been English edited by MDPI already.

8 – In the line 79 to 82, there is a lack of standardization when comparing your work regarding the genotyping codes. In the paper, nucleotide bases are used to name the genotypes, whereas the genotypes from other articles use the letters A and B. This sometimes makes confusion.

Answer-We changed the genotype in this study from G to B, as described in other articles.

9- In the line 72 to 95, significant portion of the text should be on Discussion.

Answer- The text in introduction is general discovery related to gene but details of these two specific SNPs of GH gene were shown and explained in detail in discussion.

10- In the line 104 to 106, what do the values in the parentheses mean?

Answer- We rewrote “aging between 7.1 to 68.4 months old.”

11- In the line 107, the sentence “The weight and conformation of all goats” look incomplete.

Answer-It has been English edited by MDPI already.

12- In the Material and Methods section I got a bit confused about why using data from dams and bucks and their offspring for the experiment. Why not using only the data from offspring? Did the dams and sires' mothers receive the same treatment? Why only use data from dam with two litter size kids, since the data from single births can be corrected in the model? Also, in the line 216 says that only data from 56 goats were used, how many males and females were used?

Answer: The data from offspring during the study was not enough due to the limited number of goats since this is only the pure breed BB colony in Thailand. Therefore, collection of DNA from dams and preweaning growth of dams were incorporated into this study. The diet and other management of Dams and kid diet were the same for decade.

Since number of goats were also limited, using both sex and litter size together cannot gat the results from statistic. Most of dams had 2 litter size and the statistic can come out by using these twin litter size dam.

Data of goat born from twin litter size is 56 consisted of 13 males and 43 females.

13 – In Table 2, it may be convenient to add the Taqman probes for each SNP and theirs fluorescent signals. Please correct the name of A1575G in the right column.

Answer- We already corrected and added it.

14 - In my opinion, all figures in Figure 1 could be removed. They do not add any new information.

Answer- We would like to put it in to make sure that we found “BB”.

15 – In the line 209 to 214, in my opinion, section 3.3 is not necessary since it has already been demonstrated in section 3.2 that there is no polymorphism for A1575G. It is sufficient to add this information to section 3.2."

Answer- We moved A1575G results to section 3.2

16 –In the line 217, could you improve the sentence to make it clearer or more effective? Something like: “Only sex and genes affected the growth traits.”?

Answer-It has been English edited by MDPI already.

17 – In the line 246, the word “Boar” is written incorrectly, I believe it should be “Boer”. It is repeated several times in the article.

Answer- We corrected as your suggestion.

18 – In the line 251, there are something strange in the text (Error! Bookmark not defined).

Answer: We did not see it in draft manuscript and hope it disappeared already.

19 –In lines 250 to 251, would it be possible to synthesize the text into a table and include the experiment's data as well? Maybe it will be more helpful to visualize and understand the numbers and compare them.

Answer- I would like to compare using table rather than used somebody else data to create new table.

20 – In the table 4, there are something strange in the statistical results to L10.

AA – 47.6a

AG – 43.6b

GG – 43.5ab

In this case, if different superscripts mean statistical significance among genotypes and how is it shown there is no difference between all three genotypes.

Answer: The data are corrected since the significance (0.029) means significance among groups. We have significant different between AA and AB (P=0.034) but AA & BB has P = 0.085, due to small number of BB genotype.

21 – In lines 292 to 294, it is stated that there was a sex effect on growth, but in Table 4, these differences were not statistically significant."

Answer: The significant effect of sex on H0, L0, C0 and H10 is in column of P in the title of “sex”. The last column is the P of interaction of gene*sex.

Reviewer 3 Report

Comments and Suggestions for Authors

1) Extraction of genomic DNA using blood samples is much easy than using hair follicles. Why the authors chose to use hair follicles but not blood samples?

2) The authors uses TaqMan Real-time PCR assay to genotype one SNP in one region, and used next-generation sequencing for another region. What is the reason for this? 

3)  Regarding the PCR annealing temperature, the reverse primer for A781G is very short but the annealing temperature used is 60 oC, whereas the primers for A1575G is longer but the annealing temperature is only 52 oC. can the authors explain this?

4) Figure 2 shows two double peaks in the A1575G PCR amplicon. Do these represent SNPs?

5) The sample sizes used for association study are too small. Increasing sample size is needed. 

6) SNPs need to be described using a recognised nomenclature. 

Comments on the Quality of English Language

N/A

Author Response

Reviewer # 3

Thank you very much for your suggestion especially in materials and methods. We corrected as your requests.

  1. Extraction of genomic DNA using blood samples is much easy than using hair follicles. Why the authors chose to use hair follicles but not blood samples?

Answer:  Collection of hair may be easier for training of personal to collect the samples more than blood collection. It is applicable for routine collection in breeding program. Our pilot study showed applicable extraction of DNA from hair follicle. The animal farm in this study is situated quite a distance from the provincial center, which resulted in limitations regarding access to laboratory equipment and increased transportation time. So, we collected the DNA samples from hair follicles instead of blood.

  1. The authors use TaqMan Real-time PCR assay to genotype one SNP in one region, and used next-generation sequencing for another region. What is the reason for this? 

Answer: Honestly, we initially planned to utilize the TaqMan Real-time PCR assay for both A781G and A1575G genotype analysis. However, we encountered amplification issues, possible stemming from probe sequence. Consequently, we decided to employ an alternative method, next-generation sequencing for detecting the A1575G polymorphism.

  1. Regarding the PCR annealing temperature, the reverse primer for A781G is very short but the annealing temperature used is 60 oC, whereas the primers for A1575G is longer but the annealing temperature is only 52 oC. can the authors explain this?

Answer: We determined the optimum annealing temperature using a gradient PCR assay. For the A1575G annealing temperature, optimization was conducted through a gradient PCR ranging from 52°C to 56°C. Following PCR optimization, gel electrophoresis revealed the strongest band at 52°C, which was selected as the optimal temperature for the A1575G primer.

  1. Figure 2 shows two double peaks in the A1575G PCR amplicon. Do these represent SNPs?

Answer: In Figure 2, the A1575G position is labelled in gray in the middle of figure, representing only one peak (the G base). The two peaks located in the next 10 base location are not involved in this study.

  1. The sample sizes used for association study are too small. Increasing sample size is needed. 

Answer: Since this is the only farm that raised BB goats in Thailand, we cannot increase the number of goats and we put it in limitation of the study.

  1. SNPs need to be described using a recognized nomenclature. 

Answer: We described the SNPs in this study using a recognized nomenclature (A781G and A1575G) as you recommended.

Reviewer 4 Report

Comments and Suggestions for Authors

Due to the fact that this publication is devoted to the local population of Black Bengal goats, the manuscript of the article is of certain interest.

Minor questions:

1. Line 194: for why was animals' blood taken if DNA was extracted from hair follicles?

2. Lines 195-197: You stated that the sample size was 60 females and 17 males, however, you give a litter size of 1.71, but only for dam with two kids  were included in the studies. Explanation needed!

3. Why was SNP chosen in the GH gene (A1575G) if "...has no influence on the coding potential (Line 76)"?

4. I recommend conducting an analysis of variance (ANOVA / MANOVA) with a different number of effects (gender, age) to estimate the influence of polymorphisms in the GH gene.

5. Table 4. How did the average value for trait C10 for males, which was 46.8 cm (Line 221), turn out to be higher than the average value for genotypes (41.2...44.1 cm)? Is this a typo? While the table indicates 43.1 cm!

6. I would recommend describing the history of the breeding (formation) of the studied goat population, as well as providing information about the size of this population in different countries (regions).

Author Response

Thank you very much for your valuable suggestions. We explained the points that you asked and corrected some points as your suggestion.

Due to the fact that this publication is devoted to the local population of Black Bengal goats, the manuscript of the article is of certain interest.

Minor questions:

  1. Line 194: for why was animals' blood taken if DNA was extracted from hair follicles?

Answer:  Collection of hair may be easier for training of personal to collect the samples more than blood collection. It is applicable for routine collection in breeding program. Our pilot study showed applicable extraction of DNA from hair follicle. The animal farm in this study is situated quite a distance from the provincial center, which resulted in limitations regarding access to laboratory equipment and increased transportation time. So, we collected the DNA samples from hair follicles instead of blood.

  1. Lines 195-197: You stated that the sample size was 60 females and 17 males, however, you give a litter size of 1.71, but only for dam with two kids were included in the studies. Explanation needed!

Answer: In materials and methods, we wrote “Total of 77 goats with 44 mature females, 5 mature males, 16 young females and 12 young males offsprings” which were all goats for genotypic studies in the first part of the study. The average litter size of dam (mature female) was 1.71±0.10. However, in the second part of this study on relationship with preweaning growths, we recruited all goats born from only twin litter size since litter size had strong effect on growth and birth weight.

  1. Why was SNP chosen in the GH gene (A1575G) if "...has no influence on the coding potential (Line 76)"?

Answer: Although the A1575G of GH gene does not impact the protein sequence, it may influence gene expression or regulate other genes. Several previous studies have indicated significant relationships with growth parameters when combined with different polymorphism such as A781G of the GH gene (Hua at al., 2008, Gitanjli et al., 2020).

  1. I recommend conducting an analysis of variance (ANOVA / MANOVA) with a different number of effects (gender, age) to estimate the influence of polymorphisms in the GH gene.

Answer: We tried to conduct the analysis including sex and litter size. However, since sample size is small, the results could not come out. Thus, we decided to use gender and gene. In order to eliminate the litter size effect, only goats born from twin litter size were chosen. Our statistics also used ANOVA.

  1. Table 4. How did the average value for trait C10 for males, which was 46.8 cm (Line 221), turn out to be higher than the average value for genotypes (41.2...44.1 cm)? Is this a typo? While the table indicates 43.1 cm!

Answer: Since C10 has interaction between gene and sex, the result with interaction for C10 of male in line 221 for AA and AG genotype was 46.5±1.4 and 41.8±0.8 cm, respectively. The average value of 41.2-44.1 in the table are numbers among genotypes regardless of sex. In statistical section, “the final model included the genotype (AA, AG and GG) and sex (male and female) and their interaction”.

  1. I would recommend describing the history of the breeding (formation) of the studied goat population, as well as providing information about the size of this population in different countries (regions).

Answer: We added the information of BB goats in introduction section with a specific information of our population in materials and methods.

Round 2

Reviewer 2 Report

Comments and Suggestions for Authors

The work is ACCEPTABLE. Only minor revisions are required  for publication. 

In the 3.2 topic, Is there any reason why genotyping for SNP A781G was done using the Taqman technique and for SNP A1575G the sequencing technique was used?

Line 208 – Is it possible to change the sentence for: “Only dams with two or more kids per litter were used to calculate the effect of genotype and sex on kid growth traits using the general linear model (GLM) in SAS”? In line 218, the litter size ranged from 1 to 3.

In the line 261, there are something strange in the text (Error! Bookmark not defined).

Author Response

Thank you very much for your suggestions and concern. We replied your questions as follows,

In the 3.2 topic, Is there any reason why genotyping for SNP A781G was done using the Taqman technique and for SNP A1575G the sequencing technique was used?

Answer: We originally intended to use the TaqMan Real-time PCR assay to analyze the genotypes for A781G and A1575G. However, we faced amplification challenges, likely due to probe sequence. As a result, we decided to use an alternative approach, employing next-generation sequencing to identify the A1575G polymorphism.

Line 208 – Is it possible to change the sentence for: “Only dams with two or more kids per litter were used to calculate the effect of genotype and sex on kid growth traits using the general linear model (GLM) in SAS”? In line 218, the litter size ranged from 1 to 3.

Answer – The average litter size of dam (mature female) was 1.71±0.10 as shown in line 208. Most of them came from twin litter sizes. When we determined the factors affecting the growth traits in kids, both sex and litter size had the strong effects. However, we cannot use both factors in SAS due to the limited number of goats. Therefore, we used genotype and sex for analysis in only goats that born from twin litter size. We did not use goat that born from singleton or triplet or quadrat to analyze the relationship with growth traits.

In line 261, there are something strange in the text (Error! Bookmark not defined).

Answer- I tried to look at the error but I cannot find in MS word. I removed all endnote and link already.

Reviewer 3 Report

Comments and Suggestions for Authors

One major issue is the variation in exon 4.

1) The PCR primers for this region were designed poorly. They were designed using the antisense strand of the HR gene sequence. More importantly, the reverse primer had one nucleotide deletion in the middle of primer, which could lead to poor amplification, possibly explain the need for such a lower annealing temperature.  

2) While there is not variation detected for the A1575G, and there is variation at 24 bp upstream and 10 bp downstream (refers to Figure 2). The concern is raised about why the authors did not analysis genotype data and their effect for these two SNPs.

3) The Electropherogram shown in Figure 2 looks a little odd. Has it been modified or edited?

The authors claimed that “We described the SNPs in this study using a recognized nomenclature (A781G and A1575G) as you recommended”.  What recognised nomenclature did the author use? I would recommend to use “c.xxx” to describe the SNP position.

Comments on the Quality of English Language

N/A

Author Response

Thank you very much for your valuable suggestion to improve MS.

One major issue is the variation in exon 4.

1) The PCR primers for this region were designed poorly. They were designed using the antisense strand of the HR gene sequence. More importantly, the reverse primer had one nucleotide deletion in the middle of primer, which could lead to poor amplification, possibly explaining the need for such a lower annealing temperature.  

Answer- It is possible that designed primer can lead to poor amplification. However, we identified the best annealing temperature by employing a gradient PCR assay. Specifically, for the A1575G annealing temperature, we conducted optimization across a gradient from 52°C to 56°C. Subsequent gel electrophoresis showed the strongest band at 52°C, which was determined as the optimal temperature for the A1575G primer.

2) While there is not variation detected for the A1575G, and there is variation at 24 bp upstream and 10 bp downstream (refers to Figure 2). The concern is raised about why the authors did not analysis genotype data and their effect for these two SNPs.

Answer - We analyzed the samples and found that all of them exhibit heterozygous genotypes, indicating AG at 24 bp upstream and CT at 10 bp downstream, respectively.

3) The Electropherogram shown in Figure 2 looks a little odd. Has it been modified or edited?

Answer: We modified this figure slightly by labeling the site A1575G with grey.

The authors claimed that We described the SNPs in this study using a recognized nomenclature (A781G and A1575G) as you recommended”.  What recognized nomenclature did the author use? I would recommend to use “c.xxx” to describe the SNP position.

Answer- The two SNPs in this study can be denoted as c.781A>G and c.1575A>G. We will mention these nomenclatures in the introduction section. However, for reader comprehension, we refer using them as A781G and A1575G, respectively, since it is common as shown in other articles.